# Orthobunyaviruses: From Virus Binding to Penetration into Mammalian Host Cells

**DOI:** 10.3390/v13050872

**Published:** 2021-05-10

**Authors:** Stefan Windhaber, Qilin Xin, Pierre-Yves Lozach

**Affiliations:** 1CellNetworks—Cluster of Excellence and Center for Integrative Infectious Diseases Research (CIID), Department of Infectious Diseases, Virology, University Hospital Heidelberg, 69120 Heidelberg, Germany; stefan_windhaber@arcor.de; 2Institut National de Recherche Pour l’Agriculture, l’Alimentation et l’Environnement (INRAE), Ecole Pratique des Hautes Etudes (EPHE), Viral Infections and Comparative Pathology (IVPC), UMR754-University Lyon, 69007 Lyon, France; qilin.xin@etu.univ-lyon1.fr

**Keywords:** arbovirus, Bunyamwera, cell entry, emerging virus, endocytosis, fusion, La Crosse, Oropouche, receptor, Schmallenberg

## Abstract

With over 80 members worldwide, *Orthobunyavirus* is the largest genus in the *Peribunyaviridae* family. Orthobunyaviruses (OBVs) are arthropod-borne viruses that are structurally simple, with a trisegmented, negative-sense RNA genome and only four structural proteins. OBVs are potential agents of emerging and re-emerging diseases and overall represent a global threat to both public and veterinary health. The focus of this review is on the very first steps of OBV infection in mammalian hosts, from virus binding to penetration and release of the viral genome into the cytosol. Here, we address the most current knowledge and advances regarding OBV receptors, endocytosis, and fusion.

## 1. Introduction

*Orthobunyavirus* consists of over 80 members that are globally distributed, which is a genus of the family *Peribunyaviridae* (*Bunyavirales* order) along with *Herbevirus*, *Pacuvirus*, and *Shangavirus* (Table 1) [1]. While herbeviruses and shangaviruses have a host range restricted to insects, pacuviruses and orthobunyaviruses (OBVs) infect a wide range of invertebrate and vertebrate hosts. OBVs usually spread to vertebrates, including humans and livestock, by blood-feeding arthropod vectors and therefore belong to the supergroup of arthropod-borne viruses (arboviruses) [2]. Most OBVs are transmitted by mosquitoes and culicoid flies, which are midges, and a few are transmitted by ticks and bed bugs. More than 30 members in the family are responsible for several diseases in humans. For example, Oropouche virus (OROV) is the second most frequent cause of acute but self-limiting febrile illnesses in Central America [3,4]. Ngari virus is responsible for hemorrhagic fever in Africa [5,6], and La Crosse virus (LACV) is a common cause of pediatric encephalitis in North America [7]. In domestic animals, abortion, congenital malformations in offspring, and birth defects are frequently observed following infection by some OBVs. Cache Valley virus in the United States of America (USA) and Schmallenberg virus (SBV), an orthobunyavirus that emerged in 2011 in Central Europe, are both teratogenic in livestock, predominantly in ruminants [8,9,10].

Little is known about the vector competence, prevalence, and overall burden of OBV diseases. An example is the several hundred thousand asymptomatic LACV infections that occur each year in the USA, but the actual number of infections is not known [11]. Human activity and climate change are factors that favor the spread of arthropod vectors to new geographical areas from which they were previously absent, carrying their viruses with them. Recent illustrations are the identification of Cristoli virus and Umbre virus (UMBV) in immunocompromised French patients who developed encephalitis [12,13]. UMBV was subsequently found in *Culex pipiens* mosquitoes in southern France, indicating that UMBV is now present in Europe. UMBV was first isolated in India in 1955 [14]. No vaccines or treatments against OBVs are currently approved for human use by either the Food and Drug Administration or the European Medicines Agency.

A growing body of evidence indicates that genetic reassortment between OBV members can occur in arthropod vectors [15,16]. The frequency of such events remains unknown in the wild, but it certainly contributes to the emergence of new viruses with potentially significant pathogenicity in humans. A good illustration is Ngari virus, which was found to result from the mixing of genetic material between Bunyamwera virus (BUNV) and an unidentified OBV [5]. Overall, OBVs should be seriously considered as potential agents of emerging and reemerging diseases [12,13].

In this review, we address the most current knowledge on early OBV-host cell interactions, from virus binding and uptake to intracellular trafficking and fusion. Most information on OBVs derives from a limited number of studies and an even smaller number of isolates, mainly BUNV, California encephalitis virus (CEV), LACV, and SBV (Table 1). Nothing is known about the viral penetration mechanism in arthropod cells, and hence, we limit our discussion here to the entry of OBVs into mammalian host cells. For information on OBVs and their arthropod vectors, we recommend reviewing [2,3,7,17,18,19,20,21].

## 2. OBV Genome and Viral Particles

OBVs are enveloped with a single-stranded negative-sense RNA genome that exclusively replicates in the cytosol [2]. The viral genome consists of three unique segments sharing a common organization that includes a coding region flanked by 3′ and 5′ untranslated regions (UTRs). The name of each of the three segments refers to their size in nucleotides, i.e., 6.9 kb on average for the largest segment L, 4.5 kb for the medium segment M, and 1.0 kb for the smallest segment S (Figure 1a) [2]. The terminal 3′ and 5′ ends of each segment are conserved and complementary, which allows the formation of a panhandle structure [22,23,24]. This structure is sometimes described as a hairpin and functions as a promoter for both transcription and replication [22,23,24].

The S segment codes for the nucleoprotein N and the L segment, which corresponds to the RNA-dependent RNA polymerase (RdRp) [2]. The nucleoprotein N associates with the viral genomic RNA and, together with the viral polymerase L, constitutes the pseudohelical ribonucleoprotein (RNP) structures [25,26,27]. OBVs do not have a rigid inner structure, i.e., they lack any classical matrix or capsid protein. The nucleoprotein N is therefore a key player in protecting the viral genomic information. An X-ray structure of N has been examined for many OBVs, namely, BUNV, LACV, SBV, and Leanyer virus, which has shed light on the mechanism of RNP assembly [28,29,30,31,32,33,34]. The S segment of most OBVs encodes a nonstructural protein, NSs, which is generally translated from an alternative open reading frame within the N protein-coding sequence (Figure 1a).

The M segment codes for a precursor polypeptide, the proteolytic cleavage of which by host proteases results in a second nonstructural protein, NSm, and two transmembrane envelope glycoproteins, Gn and Gc [38,39]. Proteolytic processing occurs in the endoplasmic reticulum and Golgi network (Figure 1b) [39], which are where viral particles acquire their lipid bilayer membrane and assemble. The specific location and mechanisms for the maturation of glycoproteins and virus budding may differ among cell types and remain largely undefined. As the two nonstructural proteins NSs and NSm have not been reported to contribute to OBV entry, they will not be addressed here. Studies actually suggest that NSm is involved in OBV assembly and budding [40,41]. For more details on the genomic organization, replication, and assembly of OBVs, we recommend the excellent review by Richard Elliott [2].

Electron micrographs have shown that orthobunyaviral particles are roughly spherical, with a diameter of approximately 90–110 nm [42,43]. On viral particles, the two envelope glycoproteins Gn and Gc assemble into heteromultimers and form spike-like projections of up to 20 nm (Figure 2a), which are responsible for virus attachment to the target-cell surface. Analysis of BUNV by cryo-electron tomography (cET) revealed a nonicosahedral lattice, with surface Gn and Gc protrusions exhibiting a unique tripod-like arrangement (Figure 2b) [43]. Overall, orthobunyaviral particles appear to be pleiomorphic. However, it remains unclear whether the heterogenicity in shape is an intrinsic property of OBVs or results from the sample preparation and exposition of particles to strong fixative agents prior to imaging by electron microscopy.

## 3. Receptors for OBVs in Mammalian Hosts

OBVs are mainly transmitted to mammalian hosts during the blood meal of arthropods, i.e., they are introduced into the skin dermis through bites by infected arthropods [44]. In the skin, dermal macrophages and dendritic cells (DCs) are among the first cells to encounter incoming viruses. To infect cells, OBVs and other viruses must first gain access to the intracellular environment. The very first step depends on interactions between viral particles and cell surface receptors, which can be proteins, saccharides, and lipids [45,46]. Some primary receptors can mediate the entry of viruses by themselves, whereas others rely on additional cellular cofactors or secondary receptor complexes for the entry of viral particles into host cells. We recommend the excellent reviews by Boulant and colleagues and by Maginnis for more details about virus-receptor interactions in general [45,46]. In the case of OBVs, only a few receptors and attachment factors have been reported thus far.

Virus-receptor interactions are specific, although the affinity and avidity of such interactions essentially depend on the identity of the virus and receptor. The multivalence of these interactions, through the concentration of multiple molecules of the same receptor within microdomains, often permits an increase in the avidity of low-affinity interactions [45]. The polysaccharides of glycoproteins and glycolipids are highly polar structures found on the surface of most mammalian cells, notably in the extracellular matrix. Many viruses use polysaccharides as first docking sites through electrostatic interactions, which are usually of low affinity [45,46]. Glycosaminoglycans, such as heparan sulfate proteoglycan (HSPG), have been shown to favor infection by SBV and another OBV, Akabane virus (AKAV) [47,48]. When heparinase was used to remove heparan sulfates from the cell surface, infection by SBV and Akabane OBV was impaired. This result was confirmed using cell lines with HSPG knocked out.

The OBV Germiston has been shown to target and infect dermal DCs by subverting the human C-type lectin DC-SIGN, which is also known to mediate infection by many unrelated arboviruses from different viral families [45,49]. Similarly, DC-SIGN was shown to enhance infection by rhabdoviral particles pseudotyped with LACV glycoproteins [50], indicating that the interactions between OBVs and the lectin most likely occur through the glycans carried by Gn and Gc. Whether DC-SIGN serves as an entry receptor or an attachment factor for OBVs remains to be determined. Recent studies have also identified several other C-type lectins as potential receptor candidates for LACV, namely, Mincle, Dectin-1, and Dectin-2 [51]. However, the role of these three lectins in the LACV infectious entry process remains to be clarified.

Together, C-type lectins such as DC-SIGN, Mincle, Dectin-1, and Dectin-2 represent interesting receptor candidates and may provide a molecular bridge between OBVs originating from arthropod vectors and the initial infection in the skin of mammalian hosts. DC-SIGN captures pathogens with an *N*-glycan coat composed of high-mannose residues, such as those found in the glycoproteins of viral particles derived from mosquitoes [52]. In addition, these C-type lectins are expressed on dermal macrophages and DCs, which are both present in the anatomical site of OBV transmission. However, the fact that OBVs can infect tissues and cell lines that lack the expression of these lectins indicates that these viruses can use alternative receptors to enter and infect mammalian host cells [53,54].

## 4. OBV Uptake

OBVs rely on the sorting of viral particles into the endocytic cellular machinery for infection. The transition steps between the cell surface and nascent endosomes remain largely unknown. However, depletion of cholesterol by methyl-β-cyclodextrin was shown to hamper infection by OROV and AKAV [55,56]. Cholesterol and other lipids are important for the formation of receptor-rich microdomains at the cell surface, which is a prerequisite for local curvature of the plasma membrane and receptor-mediated signal transduction, leading to virus uptake and sorting into early endosomes (EEs) [45].

The sorting of viral particles into endosomal vesicles and the endocytic route in which viruses are taken up are usually determined by amino-acid sequence motifs in the cytosolic tail of viral cellular receptors. These motifs function as docking sites for specific adaptor proteins with a role in signaling, endocytic internalization, and intracellular trafficking [45]. DC-SIGN has several such motifs in its cytosolic tail; among other factors, a dileucine (LL) motif is essential for the internalization of cargos by the lectin. DC-SIGN mutants that lack the LL motif are no longer able to internalize phleboviruses, although the viral particles bind to the mutant molecule as efficiently as to the wild-type receptor [49,57]. Phleboviruses belong to the *Phenuiviridae* family, which is closely related to *Peribunyaviridae* and hence OBVs. The importance of these signal motifs in the internalization of OBVs by DC-SIGN remains to be fully studied. No signaling motifs have actually been documented in OBV receptors with functions related to viral particle endocytosis and productive entry.

There are lines of evidence indicating that several OBVs primarily enter and infect host cells by clathrin-mediated endocytosis (CME) (Figure 3). Studies using complementary approaches aiming to impair the function of adaptor protein 2 (AP2) and clathrin chains, i.e., those based on the use of drugs, dominant negative (DN) mutants, and small interfering RNAs, highlighted the dependance of AKAV, LACV, OROV, and other OBVs on CME for infection [53,55,56]. Additional investigations into cellular receptors will be required to better characterize the uptake pathways used by OBVs. The underlying cellular mechanisms could vary considerably in different target cells. As recently discussed for phleboviruses [58], the ability of OBVs to use alternative receptors on the cell surface very likely impacts their capacity to enter one or more endocytic pathways to infect cells and tissues.

## 5. OBV Intracellular Trafficking

Upon internalization, orthobunyaviral particles proceed through the endocytic machinery until they reach the endosomal vesicles from which they fuse and enter the cytosol. Transport from EEs to late endosomes (LEs) is highly complex and dynamic, involving hundreds of cellular factors [59,60]. This process occurs along with major changes in proteins, lipids, and proton (H^+^) concentrations in endosomal vesicles [61]. The intraluminal pH decreases from ~6.5 in EEs to ~5.5–5.0 in LEs [58]. Several reports have demonstrated that OBVs depend on endosomal acidification for infection [62]. Among other factors, AKAV and OROV infections are sensitive to agents that elevate the endosomal pH, such as lysosomotropic weak bases, e.g., chloroquine and ammonium chloride, or inhibitors of vacuolar H^+^ ATPases, such as bafilomycin A1 [55,56].

Only a limited number of studies address the intracellular trafficking of orthobunyavirus within endosomal vacuoles, namely, LACV and OROV [53,55,56]. The expression of a DN mutant of endogenous Rab5, a small GTPase required for the trafficking and maturation of EEs, blocks infection by LACV [53]. OROV was found by confocal microscopy to transit through vesicles positive for EEA1 [56], a Rab5 effector protein that exclusively localizes to EEs. LACV appears to infect cells independently of active Rab7 [53], the most critical small GTPase for late endosomal maturation [63]. Nevertheless, OROV was observed entering endosomal compartments decorated by Rab7 [56].

Rab7 has been shown to be dispensable for infection by some late-penetrating viruses, a large group of viruses that depends on intact LEs for infection [64]. There can be several explanations for this outcome. As one of these explanations, viruses might simply escape the endocytic machinery earlier, for example during sorting from EEs to nascent multivesicular bodies (MVBs), an initial stage in the LE maturation process driven in part by a switch of GTPases from Rab5 to Rab7 [59]. This seems to be the case for Crimean Congo hemorrhagic fever virus, which belongs to the genus *Orthonairovirus* in the *Bunyavirales* order; hence, a genus related to *Orthobunyavirus* that has recently been shown to penetrate cells from MVBs [65]. OBVs may follow the same strategy to penetrate cells. However, little is known about the late stages of OBV intracellular trafficking, and there is overall no clear trend towards whether OBVs penetrate and infect host cells from EEs or LEs.

## 6. OBV-Cell Membrane Fusion and Penetration

The fusion of the viral envelope with cell membranes is the ultimate step in the entry program of enveloped viruses. This step allows the virus to cross the endosomal membrane, leading to the release of the viral genome and material into the cytosol. The dependence of infections on low pH suggests that OBVs achieve membrane fusion by acid activation, at least in part. It has been shown that exposure of LACV to acidic buffers in the absence of any target membrane leads to conformational changes in Gc [66]. In this respect, endosomal acidification serves as a major cue to trigger the fusion of many enveloped viruses.

The endosomal pH is sometimes not sufficient by itself to trigger virus fusion. Among other factors, proteolytic cleavage in viral envelope glycoproteins may also be required. In addition to endosomal acidification, cleavage of the glycoproteins Gn and Gc by serine proteases in target cells seems to be necessary for LACV penetration [50]. However, the cleavage sites remain to be identified, and proteolytic processing may be specific to LACV, as no other OBV has been shown to rely on the cleavage of Gn or Gc for infectious entry. At least three distinct classes of viral fusion proteins (I-III) have been documented, each with specific structural and mechanistic characteristics [67]. Forcing the glycoproteins Gn and Gc to the cell surface through overexpression has enabled the measurement of cell-cell fusion “from within” [37,68,69]. Combining this approach with site directed mutagenesis and bioinformatics analysis has led to the suggestion that the fusion unit responsible for OBV membrane fusion is carried within the Gc glycoprotein [35,36,37,68,69].

The N-terminal half of the BUNV Gc ectodomain was shown to be dispensable for Golgi trafficking and cell fusion while only the C-terminal half was required to mediate cell fusion [70,71]. These results were recently confirmed by the structural characterization of Gc fusion glycoprotein domains of several OBVs [72]. Interestingly, the N-terminal region of Gc appears to be variable overall among OBVs. In addition to the variable region of SBV Gc, Hellert and colleagues determined the crystal structures of the Gc head domains from BUNV, LACV, and OROV to resolutions ranging between 2.1 Å and 2.9 Å [72]. Interestingly, the Gc variable domain, which constitutes most of the projecting spike, appears to be the major target of the neutralizing antibody response.

The overall fold shows that the OBV Gc glycoprotein strongly resembles class II membrane fusion proteins [72]. Specific histidine residues in class II fusion proteins serve as sensors for acidification in the endosomal lumen and often define the optimal pH value for virus fusion [73,74]. Acid-activated penetration has been found to occur in a range of 5.8–6.0 for LACV and CEV [37,68,75,76,77]. However, other factors were also shown to play an important role in infectious entry. The maturation from EEs to LEs is accompanied not only by changes in the H^+^ levels but also by changes in the concentration of other ions such as potassium (K^+^) and calcium. The influx of K^+^ in LEs was shown to cause conformational changes in the matrix protein of influenza virus (IAV) and in turn, a loss of stability in viral RNPs [78]. Both the switch from sodium (Na^+^) to K^+^ and the decreased pH in LEs are needed to prime IAV cores for efficient uncoating following viral fusion.

Some OBVs were recently shown to depend on endosomal K^+^ influx. Infection with BUNV and SBV was impaired in the presence of drugs blocking K^+^ influxes [79,80]. For BUNV, it was found that K+ accumulation within endosomes is critical for infection and is regulated by cellular cholesterol abundance [81]. In contrast, a K^+^ ionophore was reported to hamper LACV infection [82]. Although the exact role of K^+^ and Na^+^ in infectious entry remains to be investigated for many OBVs, it is tempting to postulate that the late endosomal switch from Na^+^ to K^+^ has a role in OBV uncoating and release of the viral genome into the cytosol. The cell is then infected, and viral replication begins.

## 7. Conclusions and Perspectives

In this review, we have compiled the most current knowledge on early OBV-host cell interactions, from virus binding to fusion. It is highly likely that each OBV member has specificities and distinct requirements for the first steps of infection. More OBV species will have to be analyzed to identify a common trend regarding the cellular mechanisms used by these viruses to enter mammalian host cells. Nevertheless, it is apparent that OBV infections have a common dependence on intracellular vesicular trafficking and endosomal maturation and acidification, although it remains unclear whether these viruses penetrate the cytosol from EEs or LEs. It is also apparent that many aspects of the cell biology of OBV endocytosis, fusion, and penetration require further investigation. High-throughput screens involving the use of siRNAs, CRISPR/Cas9, or haploid cells, such as those performed with SBV [47] or those described for other viral families in the *Bunyavirales* order [83,84,85], will certainly help to identify new cellular factors and mechanisms important for OBV endocytosis and entry.

Ideally, preventing OBV dissemination requires approaches that target the early steps of infection. Although it is obvious that OBVs can subvert different cellular receptors to infect distinct species and tissues, only a few have been reported in mammalian hosts, and none have been reported in arthropod vectors. The identification of the host range is also paramount. Therefore, progress will imply the identification and analysis of receptors in different types of tissues in both mammals and arthropods. The combination of in vitro and in vivo studies will arguably shed light on OBV transmission, entry, and spread.

The characterization of viral particles produced from the different hosts and vectors also needs further work. The composition of proteins, lipids, and oligosaccharides in viral particles certainly influences the identity of receptors used by OBVs and hence the identity of the first-target cells as well as the subsequent sorting process into the endocytic machinery, in brief, the initial steps of infection. Viral fusion proteins and uncoating also remain insufficiently characterized, given the central role played by H^+^ and other ions in OBV penetration. Structural information on the postfusion form of the Gc protein is still missing.

A number of questions also remain regarding the arrangement and interactions of the Gn and Gc glycoproteins on the viral particles. In this respect, the cET structure of mammalian cell-derived BUNV particles is the only one available for OBVs, and it was determined at a low resolution, i.e., 3 nm [43]; an X-ray structure of a Gn ectodomain is still not available. Only a combination of crystallographic data for both Gn and Gc will make it possible to obtain a much higher resolution of the global structure of OBV particles, which in turn should lay the basis for the identification of broadly neutralizing antibodies.

These are the essential keys to broadening our knowledge of OBV dissemination and ultimately aiding the development of new antiviral strategies and approaches aiming to prevent OBV spread. To this extent, all the factors and processes that have been documented with a role in virus entry, from both the virus and cell perspectives, can potentially be targeted to block the initial steps of transmission and the subsequent spread throughout hosts.

## Figures and Tables

**Figure 1 viruses-13-00872-f001:**
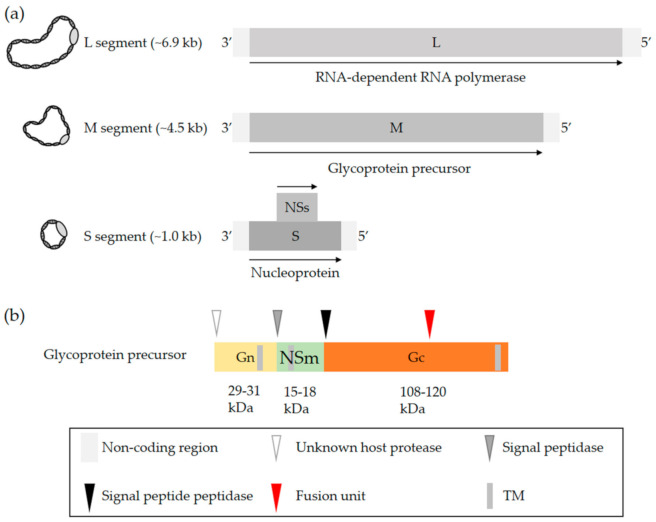
OBV genome and the glycoproteins Gn and Gc. (**a**) Generic representation of an OBV genome. The name of the three viral genomic RNA segments refers to their size, i.e., S (small), M (medium), and L (large). (**b**) Schematic representation of the OBV M precursor polypeptide. Arrow heads show the sites proteolytically cleaved by host cell proteases. The red arrowhead indicates the location of the fusion peptide based on bioinformatics predictions and biochemical analysis of LACV glycoproteins [35,36,37]. Abbreviation: TM, transmembrane domain.

**Figure 2 viruses-13-00872-f002:**
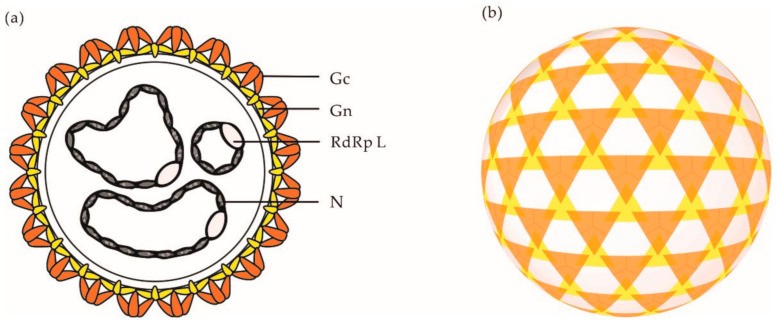
(**a**) Schematic representation of an OBV particle. (**b**) Schematic representation of the tripodal architecture of the glycoprotein spikes o the surface of a BUNV particle as analyzed by cET [43]. Abbreviations: Gn, glycoprotein Gn; Gc, glycoprotein Gc; N, nucleoprotein N; RdRp L, RNA-dependent RNA polymerase L.

**Figure 3 viruses-13-00872-f003:**
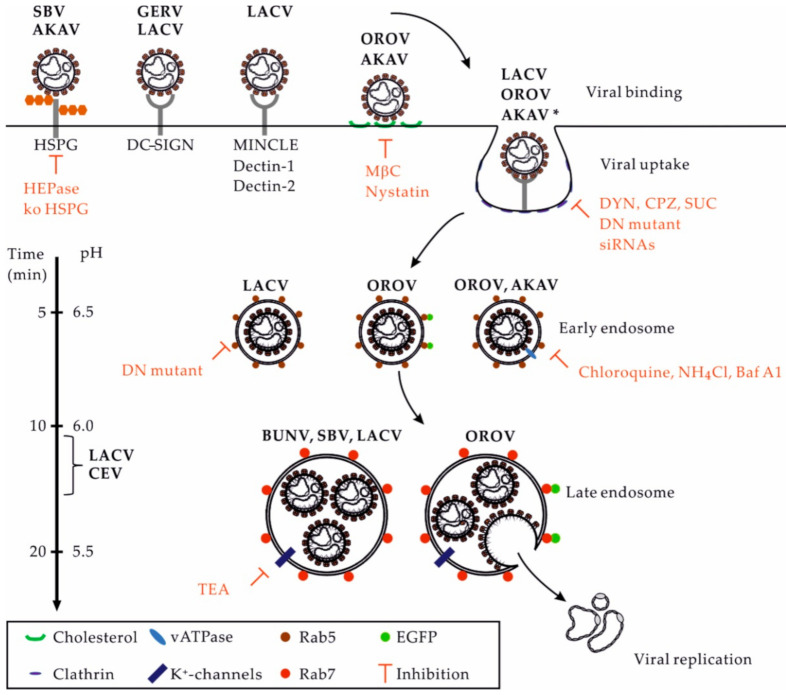
OBV entry into mammalian cells. This figure provides an overview of the entry pathways used by some OBVs to infect host cells. On the left, the scales indicate the time for the trafficking of cargo from the surface to an organelle and the related pH inside endosomal vesicles. Abbreviations: AKAV, Akabane virus; Baf A1, bafilomycin A1; BUNV: Bunyamwera virus; CEV, California encephalitis virus; CME, clathrin-mediated endocytosis; CPZ, chloropromazine; DN, dominant negative; DYN, dynasore; EGFP, enhanced green fluorescent protein; GERV, Germiston virus; HEPase, heparinase; HSPG, heparan sulfate proteoglycan; LACV, La Crosse virus; MβC, methyl-β-cyclodextrin; NH_4_Cl, ammonium chloride. OROV, Oropouche virus; SBV, Schmallenberg virus; SUC, sucrose; TEA, tetraethylammonium; * Other viruses using the CME pathway, including the CEV and Inkoo, Jamestown canyon, Keystone, Melao, Serra do Navio, snowshoe hare, Tahnya, and Trivittatus viruses.

**Table 1 viruses-13-00872-t001:** Classification within the family *Peribunyaviridae* [1].

Genus	Species	Representative Species	Hosts
*Herbevirus*	33	Herbert virus	Invertebrates
*Orthobunyavirus*	88	Bunyamwera virus, California encephalitis virus, Germiston virus, La Crosse virus, Oropouche virus, Schmallenberg virus, Umbre virus	Invertebrates, vertebrates
*Pacuvirus*	5	Pacui virus	Invertebrates, vertebrates
*Shangavirus*	1	Insect virus	Invertebrates

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
