# Peer review of "Orthobunyaviruses: From Virus Binding to Penetration into Mammalian Host Cells"

_viruses, 2021, doi:10.3390/v13050872_

Round 1
Reviewer 1 Report
Dear Dr. Lozach,
In the review manuscript entitled "Orthobunyaviruses: from virus binding to penetration into mammalian host cells" (ID: viruses-1215257),
You have provided detailed information regarding the state of the art knowledge of Orthobunyaviruses mechanisms of binding and entry into host cells.
I believe that your manuscript is well-written and easy to follow. The first steps of virus infection are clearly described and sufficiently supported by bibliographic references. In addition, you have also included nice pictures explaining virus penetration steps and how those have been elucidated by using different experimental approaches.
I believe that you paper is auseful tool to better understand the replication of emergent or re-emergent viruses as Orthobunyaviruses.
In my opinion your manuscript can be published as it stands without any revisions.
Best Regards
Reviewer
Author Response
Author response:
We acknowledge Reviewer 1 for the time spent to assess this manuscript and the enthusiastic, constructive comments about our work. This is highly appreciated.

Reviewer 2 Report
I have read this review on the entry mechanisms of orthobunyaviruses with great interest. It is well written, comprehensive and easy to read, I would like to congratulate the authors. I only have a few comments:
In the figure 1b:
1. Could the authors indicate the position of the TM segments?
2. In the legend of the figure, the authors indicated that they have localised the position of the fusion peptide based on the crystal structure of several OBs, however the crystal structure of the fusion protein is not available for any of them. As the authors explain latter in the paragraph 259-267, the crystal structure to which they refers, belongs only to the N-terminal part of Gc (head + stalk domains). I think they should remove this line from the legend, as it is misleading.
3. In the lines 97-98, the authors mentioned that some proteolytic processing occurs in the Golgi apparatus. Can they be more specific or add a reference? are they referring to a cut with furin?
4. In the lines 249-250, they mentioned a cleavage of Gn and Gc by cathepsin, could the indicate if the cleavage sites have been identified, and, if so, indicate them in the figure 1b.
5. In the figure 3. I think the authors forgot to draw the early endosomes in the middle row.
Author Response
Reviewer 2
I have read this review on the entry mechanisms of orthobunyaviruses with great interest. It is well written, comprehensive and easy to read, I would like to congratulate the authors. I only have a few comments:
Author response:
We are very grateful to Reviewer 2 for the evaluation of our manuscript and the insightful comments that, we sincerely believe, helped to improve our work.
In the figure 1b:
- Could the authors indicate the position of the TM segments?
Author response:
We have now indicated the position of the transmembrane (TM) domains in Figure 1b. The abbreviations have been updated in lines 93-94 as follows:
“Abbreviations: TM, transmembrane domain; UTR, untranslated region.”
- In the legend of the figure, the authors indicated that they have localised the position of the fusion peptide based on the crystal structure of several OBs, however the crystal structure of the fusion protein is not available for any of them. As the authors explain latter in the paragraph 259-267, the crystal structure to which they refers, belongs only to the N-terminal part of Gc (head + stalk domains). I think they should remove this line from the legend, as it is misleading.
Author response:
Reviewer 2 is correct. Therefore, we have removed the line from the legend. The order of references has been updated with Endnote to reflect the fact that the corresponding reference (number 35 in the original manuscript and now 72 in the revised manuscript) has been deleted in the legends of Figure 1. The legends for lines 91-93 now read as follows:
“The red arrowhead indicates the location of the fusion peptide based on bioinformatics predictions and biochemical analysis of LACV glycoproteins [35-37].”
- In the lines 97-98, the authors mentioned that some proteolytic processing occurs in the Golgi apparatus. Can they be more specific or add a reference? are they referring to a cut with furin?
Author response:
Several lines of evidence indicate that the M precursor is cleaved by host cell proteases in the ER-Golgi network. However, the identity of the cellular proteases and the sites of cleavage in the M precursor remain largely to be uncovered. The only work on that topic is the report by Shi et al. published in PNAS in 2016. We have now cited this reference in the corresponding sentence in lines 97-99 as follows:
“Proteolytic processing occurs in the endoplasmic reticulum and Golgi network (Figure 1b) [39], which are where viral particles acquire their lipid bilayer membrane and assemble.”
- In the lines 249-250, they mentioned a cleavage of Gn and Gc by cathepsin, could the indicate if the cleavage sites have been identified, and, if so, indicate them in the figure 1b.
Author response:
There is actually only one publication suggesting that an orthobunyavirus, namely La Crosse virus, requires endosomal proteases in target cells for infection (Hoffmann et al., 2013, J Virol). The study relied on the use of protease inhibitors and viral particles pseudotyped with the glycoproteins Gn and Gc of La Crosse virus. The authors did not identify the cleavage sites. To clarify this point, the sentence in lines 248-253 has been reworded as follows:
“In addition to endosomal acidification, cleavage of the glycoproteins Gn and Gc by serine proteases in target cells seems to be necessary for LACV penetration [50]. However, the cleavage sites remain to be identified, and proteolytic processing may be specific to LACV, as no other OBV has been shown to rely on the cleavage of Gn or Gc for infectious entry.”
- In the figure 3. I think the authors forgot to draw the early endosomes in the middle row.
Author response:
The early endosomes were in fact drawn but not much larger than the viral particles. To make it easier to distinguish between early endosomes and viral particles, we have now enlarged the size of early endosomes in Figure 3.
